# Exploring the dynamics of Shannon's information and iconicity in language processing and lexeme evolution

Alexander Kilpatrick[1]*, Rikke L. Bundgaard-Nielsen[2]

1 Center for Language Research, University of Aizu, Aizuwakamatsu, Japan. 2 School of Languages and Linguistics, University of Melbourne, Melbourne, Australia.

* alex@u-aizu.ac.jp

## Abstract

This two-part meta-study explores the relationship of *Shannon's information and iconicity* in American English, with a focus on their implications for cognitive processing and the evolution of lexemes. Part one explores the expression of information in iconic words by calculating phonemic bigram surprisal using a very large corpus of spoken American English and cross referencing it with iconicity ratings. Iconic words—those with a form/meaning resemblance—are known to be processed with a cognitive advantage, so they are included in our tests as a benchmark. Within the framework of the *Iconic Treadmill Hypothesis*, we posit that as iconic words evolve towards arbitrariness, bigram sequences become more predictable, offsetting some the cognitive costs associated with processing arbitrary words. In part 2, we extend *Cognitive Load Theory* and the *Lossy Context Surprisal Model*—both sentence level language processing models—to test our predictions at the bigram level using the results of a battery of existing psycholinguistic experiments. In line with these models that explain the psycholinguistic consequences of hearing improbable words in sentences, our results show that words made up of improbable phonemes are processed with cognitive disadvantage, but that extra processing effort enhances their retention in long term memory. Overall, our findings speak to the cognitive limitations of language processing and how these limitations influence lexeme evolution.

## 1. Introduction

Languages are often assumed to be structured in ways that facilitate efficient and fast communication for both the speaker and the listener [e.g., 1–5]. This principle underpins linguistic theories suggesting that human speech operates near cognitive and articulatory limits to optimize communication [5]. This report presents a two-part meta-study testing this assumption through the lens of phonemic bigram surprisal—a measure of the unexpectedness of one phoneme following another—and

**Data availability statement:** All data and code relating to this project can be found here: https://osf.io/mduc9/?view_only=e20f9f4d92d-54044b0ab941122b3d24a

**Funding:** AK - GRANT_NUMBER: 20K13055 Japan Society for the Promotion of Science (Tokyo, JP). https://www.jsps.go.jp/english/ The funders played no role in this study.

**Competing interests:** The authors have declared that no competing interests exist.

its relationship with iconicity, the degree to which a word's form is connected to its meaning. For example, the word buzz replicates the sound produced by a vibrating insect. Iconic words like buzz are advantageous in cognitive processing, as their form-meaning congruence enhances recognition and recall [6–10]. The first part of this study calculates phonemic bigram surprisal across a lexicon and explores its interaction with iconicity and word length. While high-surprisal events and stimuli convey more probabilistic information, they are often harder to process [11]. Cross-linguistically, words consisting of low-probability phonemic sequences convey more probabilistic information than words with high-probability phonemic sequences, and as words grow longer—or as the number of potential word competitors fall away—they express less information [12]. This study hypothesizes that iconic words defy this trend, maintaining high surprisal despite their length due to their relative ease of processing.

In the second part of this paper, we explore how surprisal and iconicity influence key psycholinguistic tasks, including auditory lexical decision tasks, reading aloud tasks, memory processing, and developmental trajectories for lexemes using a battery of existing psycholinguistic experiments. These experiments test the *cognitive economy principle* which posits that language evolution optimizes the balance between cognitive processing demands and communicative effectiveness. By analyzing the impact of surprisal and iconicity across these tasks, we aim to illuminate how language adapts to cognitive constraints, potentially shaping its evolutionary trajectory.

If languages are structured to facilitate efficient and fast communication [5], one might expect human speech to approach the limits of articulatory production or listener processing capabilities. This is not the case, and it raises the question of why and how speakers settle on relatively stable and language-specific speaking rates—much slower than what is possible to produce and comprehend. In any language, there is some variability in how quickly people speak in regular conversation; however, there is also considerable variation in the average speech rate *across* languages. For example, English is a relatively slow language in which speakers produce only 6.19 syllables per second, while Japanese speakers manage 7.84 syllables per second [13]. Speech rate appears to be constrained by how much information is packed into the syllables of a given language [14]. This is presumably due to cognitive constraints: Cross-linguistic comparisons indicate that languages that express more information per syllable are spoken at a slower rate leading to languages converging in how much information they express over time [14].

How much information is transmitted per second is often investigated from an Information Theoretical [15] approach. Here, *events* (e.g., words, syllables, or phoneme sequences) with low likelihood are considered to convey more information compared to highly expected or predictable events. Predictability within this framework is quantified using the surprisal equation, which is a log-transformed expression of probability that returns values in bits of information. Phonological surprisal is calculated on the transitional probability of phonemes co-occurring in a language. English has a larger phonemic inventory than Japanese and many more

phonotactic possibilities because Japanese adheres to a strict set of phonotactic rules (although see [16]). In languages like English, which possess more potential sound combinations comparatively, this reduces the overall transitional probability of sequences of sounds resulting in an increased average expression of information per segment or syllable relative to languages like Japanese.

Another factor found to influence linguistic processing is whether words are *iconic* or *arbitrary*. Iconic words—those with a close relationship between sound and meaning—have been found to be processed faster than arbitrary words in visual lexical decision tasks [9] suggesting decreased cognitive load in their processing compared to arbitrary words. Onomatopoeia exemplifies this phenomenon, representing words that phonetically imitate or resemble the sounds associated with the objects or actions they describe, like the word buzz, discussed above. Iconic words are well-known to be important in child language acquisition and are typically acquired at an early age [e.g., 17]. The special status of onomatopoeia is not confined to child language acquisition; however. Research shows that iconicity has facilitatory effects in lexical processing, demonstrating the benefits of iconic mappings beyond language learning and suggest continued influence in adult everyday language use [e.g., 10]. These findings align with the idea that iconicity can make language more direct and vivid [e.g., 8] and may even play a role in the evolution of language [18,19]. Overall, the presence of iconicity in words appears to be a powerful tool for facilitating language comprehension and cognitive efficiency.

The present study tests the central hypothesis that surprisal and iconicity shape the allocation of cognitive resources during language processing and can influence the evolution of lexemes. Specifically, our findings suggest that as the amount of information conveyed increases, there is a corresponding impact on the level of attention required and the cognitive demands placed on individuals engaged in language-related tasks. Conversely, iconic words, due to their inherent connection between form and meaning, are processed with greater ease. In addressing these challenges, individuals may adapt their processing strategies, reallocating resources to accommodate varying levels of surprisal and iconicity, ultimately facilitating efficient language comprehension and production. In Section 1.1, we discuss Shannon's Surprisal and explain how it is calculated here; In Section 1.2, we discuss iconicity—the relationship between form and meaning—and its known influence on processing; and finally Section 1.3 considers a cognitive economy principle at play that might relate to our findings.

## 1.1 Surprisal

Information theory is a branch of mathematics focused on the quantification, storage, and transmission of information and plays a vital role in understanding and optimizing communication systems. In information theory, two mathematical tools, Shannon's *surprisal* and *entropy* [15], are used to quantify and analyse the uncertainty, predictability, and information content of events and data. Surprisal is the negative logarithm of the probability of an event ($P$), base 2. The more unlikely an event, the higher the surprisal which is expressed in values of bits of information. For example, any outcome of a fair coin toss ($P = 50\%$) carries 1 bit of information while any outcome of a six-sided die roll ($P = 16.67\%$) carries 2.585 bits.

$$Surprisal = -\log_2 P$$

Entropy serves as a measurement for assessing the degree of uncertainty or randomness within a system or dataset. It essentially provides a quantification of the system's unpredictability or disorderliness. Entropy ($H$) is calculated as the sum of the probabilities of all events multiplied by their surprisal within a given context. High entropy reflects heightened unpredictability and randomness, while low entropy indicates a greater degree of predictability and order. For instance, the entropy for both a coin toss and a die roll are identical ($H = 1$) because each outcome is equally probable. However, if we extend this to tossing two coins ($H = 1.5$) or rolling two dice ($H = 3.274$), entropy differs due to variations in the distribution of outcomes. In other words, entropy increases as the complexity of the event's outcome distribution increases.

$$Entropy = -\sum i \, (Pi \times \log_2 Pi)$$

Zipf's law of abbreviation [1,20,21]—which posits that more frequent words tend to be shorter than less frequent words—is a robust crosslinguistic phenomenon [22]. Equally robust is Zipf's law of frequency (often abbreviated as Zipf's Law; [1]), which states that the rank of a word is inversely proportional to its frequency. In any natural corpus, Zipf's law of frequency reveals a logarithmic relationship, indicating that human languages tend to have a few words with disproportionately high frequencies, while the vast majority occur much less frequently. Taken together, Zipf's law of abbreviation and Zipf's law of frequency indicate that the evolution of human language is geared towards communicative efficiency—favouring concise forms for frequently used expressions (for a review of how efficiency shapes language, see [5]). In short, languages are structured to facilitate robust, fast, and efficient communication for both the speaker and the listener [e.g., 1,2,4]. Despite the extreme distribution skew in natural corpora, shorter words tend to be constructed from less probable sequences of phonemes [22]. Less-probable words are composed of segments that convey more disambiguating information at the start of the word [12], which is essential for efficient word recognition. In another way, as words grow longer, surprisal decreases as the number of potential competitors decreases. In the present study, we calculate average bigram surprisal (hereafter: average surprisal), which refers to mean information in a word, i.e., the sum ($\sum$) of each surprisal value in each word ($-log_2Pi$) divided by the number of bigrams ($n$). A similar calculation was applied to entropy. However, because entropy was not particularly informative and is correlated with surprisal, we excluded that analysis for clarity and brevity.

$$Average\ Surprisal = \frac{\sum -log_2Pi}{n}$$

Transitional probability—including surprisal and entropy—has been used to forecast and explain a range of linguistic behaviours. As noted above, Coupé et al. [14] measured the surprisal of sequences of speech sounds against the speech rate of speakers from 17 languages. Interestingly, they found that those languages that express more syllables over time typically carry less information per syllable and vice versa. They showed that languages trend towards encoding similar information rates (39 bits/s) and proposed that this may be evidence that probabilistic information transmission is modulated by shared processing limitations.

Transitional probability also influences speech rate through probabilistic reduction which is the observation that linguistic elements, including segments, syllables, and words, characterized by higher transitional probability, tend to be produced with reduced articulatory effort such as reduced duration, increased lenition, and more centralised vowels [23–29]. Some have suggested that probabilistic reduction reallocates attentional or cognitive resources, leading to a trade-off with complexity in the acoustic signal [30]. In line with this, the present study proposes a perceptual corollary to this behaviour.

Predictability also influences speech perception such that word frequency [e.g., 31,32] and transitional probability [33–36] can tip the scales in perceptually ambiguous contexts. The interplay between probabilistic reduction and probabilistically motivated perception might be summarised as human perception makes assumptions based on existing probability information, rather than attending to acoustic signal, in contexts where speakers invest less articulatory effort. This relationship provides another viewpoint of Zipf's [1] principle of least effort which states that speakers will only invest as much time and energy as necessary to communicate effectively because it suggests that listeners reserve cognitive resources when attending to predictable stimuli.

Lindblom's [3] *Hypo- and Hyper-speech* (H&H) theory provides a foundational framework for understanding how speakers and listeners dynamically negotiate phonetic variation to balance efficiency and clarity in communication. H&H theory aims to explain phonetic variation in speech communication by comparing it to a *tug-of-war* between the speaker and listener, where the speaker strives for economy and reduced articulatory effort, while the listener requires sufficient phonetic information to understand the message. The theory suggests that speech reduction is an adaptive compromise between the speaker's desire for efficiency and the listener's need for clarity, influenced by factors such as speaker physiology, acoustics, and listener expectations. The theory has been extended by others, such as Aylett and Turk [23], who incorporated the idea of signal and language redundancy to explain how speakers manage phonetic reduction.

The present study builds on these foundations by exploring how phonemic surprisal and iconicity interact to influence cognitive processing and lexeme evolution, extending the scope of H&H theory beyond phonetic variation to include psycholinguistic and information-theoretic dimensions. By analyzing surprisal at the bigram level, it examines how iconic words—those with a form-meaning resemblance—resist typical patterns of reduction due to their inherent processing advantages. This approach highlights how linguistic structures adapt to balance cognitive effort and communicative efficiency, revealing how surprisal and iconicity jointly shape memory retention, reaction times, and developmental trajectories in language. Through this lens, the study refines H&H framework, demonstrating its relevance to broader aspects of language processing, acquisition, and evolution.

## 1.2  Iconicity

Iconicity—the correspondence between linguistic form and meaning—is evident across various sensory dimensions in human language (cf. [37], for extensive discussion), including size (e.g., [38]), shape (e.g., [39]), colour ([40]), and other sensory perceptions [41]. But iconic words do not necessarily remain iconic over time. *The Iconic Treadmill Hypothesis* [42] proposes that over very long periods of time, words tend to lose their iconicity in a process of de-iconization, potentially giving rise to more predictable phonemic sequences as part of that transformation. Indeed, research suggests that words in earlier, more iconic stages, express more Shannon's information than those in later, more arbitrary stages [43]. The evolution of the English word laugh is a notable example. In Old English, it was *hlehhan* [44], where the form-meaning resemblance is clearer. Although we have no spoken corpus to draw upon, the initial hl- cluster was likely uncommon—or surprising—in Old English because it has not survived. Over time, as the word evolved and its iconic properties diminished, it transformed into the more phonotactically common and predictable CVC sequence we recognize today (see Table 1 for additional examples).

Alongside losing iconic associations, de-iconization can also reduce marked phonological traits [45] such as the use of rare or non-inventory speech sounds (e.g., *ugh* [əx]), phonotactic violations (e.g., *vroom* [vɹum]), expressive gemination (e.g., *KAP-POW!* [kəpːaʊ]), vowel lengthening (e.g., "Nooo!" [noː]), and expressive metathesis (e.g., *brid* [bɹɝd] from bird). In the present study, we investigate whether phonemic bigram surprisal serves as a reflection of phonological markedness in iconic words. By analyzing how the predictability of phonemic sequences correlates with iconicity, we aim to elucidate the relationship between iconic forms and their phonological characteristics. Our findings will contribute to understanding the cognitive mechanisms underpinning the processing of iconic words within the broader framework of lexeme evolution.

Research has documented that iconicity plays a role in language development, cognition, and processing efficiency, highlighting its significance in linguistic comprehension and acquisition [17,46–48]. For instance, a longitudinal investigation demonstrated a tendency for both children and parents to employ a greater number of iconic words during the earlier stages of childhood, with a transition towards utilizing more arbitrary language occurring concurrently with increased age which is presumed to reflect cognitive development [17]. Iconic words are processed faster and more effortlessly than arbitrary words, possibly due their ability to evoke sensations resembling those that they imitate [49], or because of enhanced connections between phonology and semantics [9,50], or even possibly because they better evoke episodic memories in addition to declarative knowledge [10].

**Table 1. Examples of words that have undergone the deiconization process taken from an etymological dictionary [44].**

| Old English | Middle English | Modern English |
|---|---|---|
| hlahhan [ˈhlaxːan] | laghen [ˈlaxən] | laugh [lɑːf] |
| gnagan [ˈgnaɣan] | gnaʒen [ˈgnawən] | gnaw [nɔː] |
| fneosan [ˈfneozan] | nesen [ˈneːzən]/ snesen [ˈsneːzən] | sneeze [sniːz] |

Like is the case for the *surprisal* of words in sentences, previous studies have examined the influence of *iconicity* on auditory lexical decisions, reading speed, and memory recognition. For instance, onomatopoeic words elicited a smaller N400 in an electroencephalographic auditory lexical decision task, indicating facilitated lexical access, although this did not translate into differences in processing speed [51]. In terms of reading speed, words with higher iconicity were processed faster in a visual lexical decision task, suggesting that despite being presented visually, iconic links between sound and meaning can make these words easier to process and potentially increase reading speed [10]. The same study [10] also suggests that iconic words, by imitating sensory experiences, could lead to a richer encoding experience and improved retrieval, by establishing a connection between elements of form and meaning based on resemblance, which simplifies the process of learning iconic words (for a review, see [50]). This is because the perceptual similarities inherent in iconic words can serve as a scaffold for learners, aiding them in grasping the concept that words can represent real-world objects or actions. These findings collectively underscore the facilitatory role of iconicity in language processing, acquisition, and its potential to enhance cognitive functions related to language comprehension and memory.

While the cognitive advantages of iconic words are well-established, it is crucial to explore their communicative utility as well. Some researchers [52] underscore the significance of onomatopoeia in early language development, illustrating how iconic forms facilitate interactions between children and their caregivers. Others [53] similarly demonstrate that ideophones transcend mere efficiency, enhancing expressive communication across various languages. Furthermore, iconic gestures [54] create imagistic renditions that convey meaning more effectively than arbitrary language alone when combined with iconic words. Although previous research has often emphasized how iconicity aids word learning, Nielsen and Dingemanse [50] advocate for a broader understanding of its role in language. They contend that iconicity is not merely about easing processing or learning; it encompasses a wider array of functions in communication and language evolution. Our findings, particularly regarding the patterns of surprisal in iconic words, bolster this expanded perspective, highlighting how iconicity interacts with cognitive processing while fulfilling essential communicative roles.

In the present study, we explore the relationship between surprisal and iconicity which extends existing work [55] which explored the relationship between iconicity and structural markedness, incorporating phonotactic measures such as log letter frequency, phonological density, biphone probability, and triphone probability. Their findings suggest that iconic words often exhibit phonotactic distinctiveness, which may serve as a metacommunicative signal, reinforcing their performative and playful characteristics. While our study primarily investigates phonemic bigram surprisal as a measure of information content in iconic words, these phonotactic measures are conceptually related. Specifically, phonemic surprisal and phonotactic probability both capture the degree of predictability within a linguistic system, though at different levels of granularity. Additionally, other studies [56] report that log letter frequency correlates with iconicity ratings, further supporting the idea that iconic words tend to be structurally marked. Future research could explore whether phonemic surprisal and phonotactic probability jointly contribute to the cognitive processing advantages observed for iconic words.

### 1.3 Cognitive economy principle

Language processing is shaped by both predictability and the efficient allocation of cognitive resources, as the brain minimizes effort while maximizing comprehension. In the following study, we examine the interplay between surprisal and iconicity, as well as the results of an auditory lexical decision task, a reading aloud task, a memory recognition task, and two age of acquisition experiments. While this specific intersection may not have been directly studied before, the concept of predictability and its impact on language processing has been well-documented, specifically at the word- and sentence-level. For instance, [57] reveals that words which are highly predictable by their preceding context are read more quickly than unpredictable words. Indeed, decades of experimental work in expectation-based approaches to language processing (see for instance [58–60]) have shown that listeners build context-based expectations about upcoming linguistic input at different levels. These studies have revealed that predictable words are read faster, fixated on for shorter periods of time, and are more likely to be skipped than unpredictable words (e.g., [61,62]).

The *cognitive load theory* [11] posits that the human cognitive system has a limited capacity for processing information, which necessitates the efficient management of cognitive resources. The findings from expectation-based language processing research align with this theory, suggesting that predictability serves as a cognitive economizer. When words are predictable within a given context, they require less cognitive effort to process, thereby conserving the cognitive resources of the reader or listener. This conservation of cognitive resources is reflected in the faster reading times and reduced fixation durations for predictable words, as the cognitive system is not overburdened by the need to integrate unexpected linguistic input. Conversely, unpredictable words demand a greater share of cognitive resources to reconcile with the context, resulting in slower processing and increased cognitive load. Thus, the principle of cognitive economy is evident in language processing, where the brain optimizes its limited cognitive resources by favouring predictability and minimizing the effort required to process linguistic information.

The *lossy-context surprisal model* [63] extends this by accounting for the imperfect nature of memory representations in language comprehension. This model suggests that the difficulty in processing a word is directly proportional to its surprisal value, given a memory representation that may be degraded or 'lossy'. In essence, as the predictability of a word decreases, the surprisal increases, necessitating a higher cognitive load to integrate this less expected word into the current context. This increased cognitive effort can lead to slower processing times and reduced accuracy; however, the model also posits a potential benefit to this increased effort: the additional cognitive resources expended on processing high-surprisal words may result in deeper encoding and better memory recognition. This implies that while unpredictability may initially pose a challenge to comprehension, it could ultimately enhance long-term retention of information, thereby contributing to a more robust linguistic memory.

While the cognitive load theory and the lossy-context surprisal model have traditionally been explored at the word- and sentence-level, the study at hand investigates these concepts at the more granular bigram-level. This novel approach allows for a detailed analysis of how each pair of adjacent phonemes or letters within individual words contribute to cognitive load and surprisal. By focusing on bigrams, the study aims to uncover the subtleties of language processing that occur at this fundamental level of linguistic structure, providing insights into the incremental nature of comprehension and the cognitive mechanisms that underlie the integration of each successive unit of language. This finer resolution of analysis may reveal new patterns of predictability and surprisal that influence cognitive processing in ways not previously understood through larger unit investigations. The concept of cognitive economy is central to this exploration, as it posits that the human cognitive system is optimized to reduce the cognitive effort required for language processing. By examining bigrams, the study seeks to understand how cognitive resources are allocated and conserved in the face of varying levels of predictability and surprisal, and how these allocations affect the overall efficiency of language comprehension and memory recognition. This could provide a more nuanced understanding of the cognitive economy principle, showing how the brain economizes its processing efforts even at the most basic levels of linguistic analysis.

## 2. Method

All data and code relating to this project can be found here: https://osf.io/mduc9/?view_only=e20f9f4d92d54044b0ab941122b3d24a

In this study, we employed a multifaceted approach to investigate the influence of various linguistic factors on surprisal and iconicity and their impact on language processing and acquisition. We first took a very large corpus of spoken American English [64] and cross-referenced this material to a pronouncing dictionary [65] to retrieve a phonemic transcription of each word, which were then used to calculate surprisal. We obtained iconicity ratings by cross-referencing an existing study [56] in which words were rated according to their iconic relationship to their meaning. The average surprisal and iconicity values—among others—are then measured against the results of a battery of psycholinguistic experiments to test the hypotheses below.

## 2.1 Surprisal and iconicity

The initial dataset was obtained from the SUBLEX-US corpus [64] consisting of approximately 50,000,000 words extracted from subtitle data from spoken English. This dataset was cross-referenced with the Carnegie Mellon University Pronouncing Dictionary (CMU: [65]) to convert English orthography to phonemic transcriptions based upon Standard American English pronunciation. All words in the database which did not match a word entry in the CMU were discarded, as was any word consisting of a single phoneme. So, for example, "an" is included in the corpus while "a" is excluded because we calculate bigram surprisal (i.e., the probability of /a/ being followed by /n/, for instance) which requires more than one phoneme. Due to the format of the data, surprisal was only calculated within words and no consideration was given to word boundaries. Parts of speech [66] and morpheme counts [67] were included in the master dataset by cross-referencing existing datasets. The distribution of words to parts of speech as well as average surprisal, average iconicity, and average phonemic length are presented in Table 2, which shows that there are many more nouns, adjectives, and verbs than other word classes. Iconicity scores were extracted from an existing dataset [56] in which 1400 American English speakers rated how similar each word "sounds like" its meaning on a 7-point Likert scale. In the master dataset, any word that did not find a match across all additional datasets mentioned thus far was discarded resulting in a dataset of 39,136,598 instances of 13,336 unique words. All statistical hypothesis tests were constructed in R [68].

Using this dataset, we test whether iconic words exhibit higher surprisal values and if the inverse relationship between length and average surprisal will be dampened in iconic words. As noted above, as words grow longer, they become more predictable. This predictability facilitates faster processing and easier integration into the mental lexicon, thus serving the cognitive economy by reducing the effort required during language comprehension. However, iconic words may resist this trend due to their relatively effortless processing. If accurate, the effect of word length on average surprisal should be attenuated in iconic words.

## 2.2 The psycholinguistic battery

The first of the psycholinguistic datasets consists of responses to an Auditory Lexical Decision test [69], involving 231 American English-speaking participants who made lexical decisions based on auditory stimuli from the Massive Auditory Lexical Decision (MALD) dataset. The MALD dataset contains both real words and nonce words but given that the Master Dataset consists exclusively of words also found in the CMU, our analysis excludes the nonce word response data. Reaction times and accuracy (percentage of real words classified as real words) responses were recorded. A total of 10,340

Table 2. Summary of word class characteristics including count, average surprisal, average iconicity ratings, and average length measured by phoneme count.

| Class | Count | Surprisal | Iconicity | Length |
|---|---|---|---|---|
| Noun | 7270 | 4.31 | 3.74 | 5.86 |
| Adjective | 2808 | 4.15 | 3.79 | 6.96 |
| Verb | 2641 | 4.17 | 3.97 | 5.48 |
| Adverb | 210 | 3.98 | 3.32 | 5.62 |
| Name | 171 | 4.3 | 3.8 | 4.41 |
| Preposition | 52 | 3.74 | 3.18 | 4.67 |
| Pronoun | 42 | 3.5 | 3.52 | 4.1 |
| Number | 34 | 3.86 | 3.56 | 4.38 |
| Interjection | 29 | 5.58 | 5.2 | 3.1 |
| Determiner | 24 | 3.65 | 2.88 | 3.42 |
| Conjunction | 20 | 3.84 | 2.67 | 3.95 |
| Article | 3 | 2.1 | 2.47 | 2.67 |

samples from the MALD dataset match the master dataset. Almost all real words in the dataset were correctly recognised all the time ($M = 94.7\%$, $SD = 12.7\%$), and 8515 real words achieved perfect recognition accuracy (100%). Importantly, however, the distribution of response time statistics follows a bell curve-like pattern ($M = 879$ ms, $SD = 152$ ms), indicating a more typical and informative distribution for statistical models.

The second additional dataset consists of data from a Speeded Reading Experiment [70]. The speeded reading experiment—a part of the English Lexicon Project (ELP)—was designed to measure the speed and accuracy of word recognition. 816 native English-speaking participants were recruited from six different universities in the Midwest, Northeast, and Southeast regions of the United States. Participants were presented with words on a computer screen and instructed to read them aloud as quickly and accurately as possible. Each participant contributed approximately 2,500 responses, and the sessions were conducted over two different days, with no more than one week between sessions. Participants' vocal responses were recorded, and they also manually coded the accuracy of their pronunciation. Of the 13,109 words in the ELP dataset that found a match in the Master Dataset, 8450 achieved perfect accuracy ($M = 97.2\%$, $SD = 5.6\%$), although there was more variation in the response time data ($M = 474.7$ ms, $SD = 79.2$ ms).

We include two datasets from studies that examine the Age-of-Acquisition of words. The first study [71] asked 1960 American English participants to indicate the age at which they thought that they would have understood the meaning of a given word if somebody had used it in front of them. This dataset provided 12,465 words that were also included in the Master Dataset ($M = 9.22$, $SD = 2.72$). Participants were recruited from the United States using an online crowdsourcing marketplace. The second Age-of-Acquisition dataset [72] included responses from 829 participants recruited from the University of Glasgow. Each participant rated 100–150 words according to when they estimated they learned the word. 4101 words in the Glasgow norms found a match in the master dataset ($M = 4.11$, $SD = 1.2$).

Lastly, we examine a memory recognition study [73] which involved 120 undergraduate students who attended two 1.5 to 2.0 hour-long sessions in a week. In the first session, participants studied and were tested on their recognition of lists of words, while the second session involved a computer-based test where participants identified words as "old" or "new" based on their first session. The study's results were reported as successful recognition minus false alarms, yielding a mean accuracy of 72% ($SD = 20\%$) and a mean false alarm rate of 20% ($SD = 8\%$). Despite the high accuracy, no sample achieved perfect recognition accuracy, with an average recognition accuracy of 53.5% ($SD = 13.1\%$). Unfortunately, no response time data was available for this experiment.

By combining these additional datasets with the master dataset, we test whether high average surprisal words are processed with a cognitive disadvantage, leading to longer reaction times and lower accuracy in the auditory lexical decision and speeded reading datasets. We also predict that they will be acquired later according to the results of the age of acquisition tasks, reflecting their relative difficulty in early language learning. While we predict that decreased accuracy will be associated with high surprisal in the auditory lexical decision and speeded reading datasets, we predict the opposite pattern for the memory recognition experiment.

## 3. Results

### 3.1 Surprisal and iconicity

**3.1.1 Surprisal and word position.** Prior to exploring the relationship between surprisal and iconicity, we were interested in examining how information is expressed across words. To explore how bigram position in word influences surprisal, we generated heatmaps to visualize the distribution of information across the different positions according to phonemic length (see Fig 1). Fig 1 focuses on words with fewer than 11 phonemes due to the limited number of longer words in the master dataset. The heatmap reveals a distinct pattern indicating that word length predominantly affects information expression at the beginning and end of words, with increased surprisal observed from the second bigram to the third-to-last bigram in words that are 5 phonemes (i.e., 4 bigrams) or longer. This observation may be attributed to the presence of English affixes, which are high frequency sequences typically found at English word boundaries. To

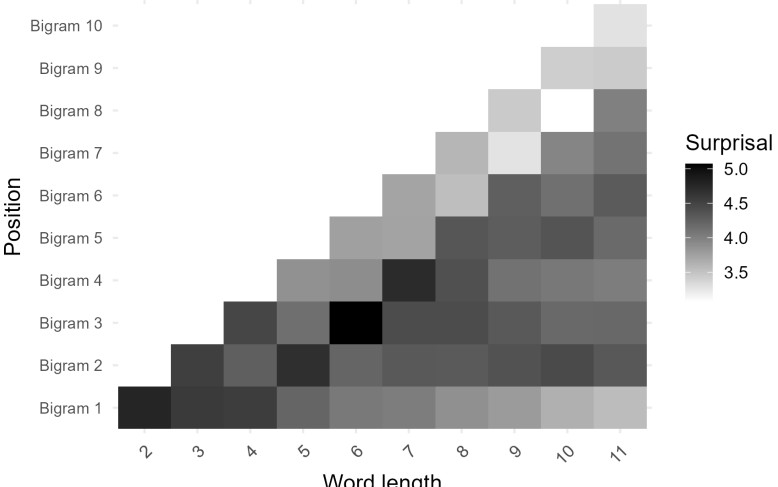

**Fig 1. Heatmap illustrating the distribution of information across bigram positions for words with 11 or fewer phonemes.**

investigate further, a second heatmap was generated using only single-morpheme words, eliminating any samples that might contain an affix. The resulting heatmap (Fig 2) mirrors the pattern observed in Fig 1 suggesting that decreased surprisal at the boundaries of longer words is not entirely modulated by affixes.

To further explore the relationship between surprisal and word position, we conducted three simple linear regression analyses on bigram surprisal at the first (Word Start), mid (Word Middle), and final (Word End) bigrams in each sample against word length. In samples with an even number of bigrams, the average of the middle two was taken for the Word Middle variable. The Word Start model, $F(1, 13302) = 454.8$, $\beta = -0.139$, $p < .001$, $R^2 = 0.033$, indicated that longer words tended to be less informative at their onset. Similarly, the Word End model revealed that the final bigrams in words express information according to length, $F(1, 13302) = 596.7$, $\beta = -0.171$, $p < .001$, $R^2 = 0.043$, emphasizing the importance of both word beginnings and endings in information expression. However, the Word Middle model exhibited a much smaller effect size, $F(1, 13302) = 4.913$, $\beta = -0.017$, $p = .027$, $R^2 < 0.001$, suggesting a weaker, albeit significant, influence of word length on surprisal in the middle of words. However, it is important to note that this model includes 1 and 2 bigram samples where Word middle is the same as Word Start and Word End and likely the reason for any observable effect in the Word Middle model. These patterns held when the models only included monomorphemic samples where the Word Start ($F(1, 6155) = 127.5$, $\beta = -0.156$, $p < .001$, $R^2 = 0.02$) and Word End ($F(1, 6155) = 193$, $\beta = -0.213$, $p < .001$, $R^2 = 0.03$) models revealed a stronger influence on surprisal than the Word Middle model ($F(1, 6155) = 6.9$, $\beta = -0.037$, $p = 0.008$, $R^2 = 0.001$). In conclusion, the examination reveals that the effect of length on surprisal is most apparent from the contrast between the middle of words and their boundaries while the significant regression equation observed in the Word Middle models are likely observable because they include samples that are 2–3 phonemes long which pick up the effects of word boundaries.

**3.1.2 The interaction between surprisal and iconicity.** To explore the relationship between surprisal and iconicity, we constructed a multiple linear regression model, assessing the effect of variables 'Parts of Speech' (PoS), 'Number of Morphemes' (n_morph), 'Phoneme Count' (length), 'Iconicity', and the 'Interaction between length and iconicity' (length:iconicity). The results presented in Table 3 indicate that length, measured by phoneme count, is not associated with decreased surprisal directly, but rather through the interaction of length and iconicity. This suggests that as words grow longer, there is a decrease in information expression, but that iconic words are resistant to this effect. In another way, words with greater iconicity tend to convey more information per phoneme. Moreover, the inclusion of morphological

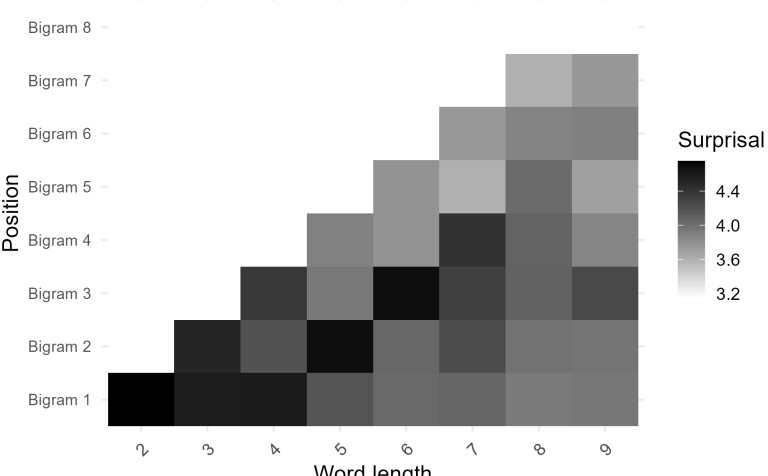

**Fig 2. Heatmap depicting the distribution of information across bigram positions for single-morpheme words with 9 or fewer phonemes.**

complexity, represented by the number of morphemes (n_morph), reveals a positive relationship with surprisal. Specifically, as the number of morphemes increases, so does the surprisal of the word provided phonemic word length is taken into consideration. Additionally, specific parts of speech stand out for their high levels of surprisal. Nouns in particular exhibit consistently high coefficients across models as do interjections, though we note that there are only 29 interjections included in the Master Dataset.

### 3.2 The psycholinguistic battery results

Having demonstrated the relationships between various linguistic factors and surprisal in the master dataset, we now extend our investigation to the battery of psycholinguistic experiments. In these experiments, we explore how average surprisal, iconicity, word length, number of morphemes, and parts of speech influence the accuracy and reaction time results of an auditory lexical decision task, a reading aloud task, a memory recognition task and two age of acquisition experiments. We test the predictions presented above by constructing a series of models with the accuracy, response time, and acquisition results as the dependent variables. These models consist of all the same independent variables as the model presented in Table 3 except for the inclusion of average surprisal as an independent variable and the exclusion of the interaction effect of length on iconicity. Due to the data distribution, the lexical decision and reading task accuracy results are tested with quasibinomial logistic regression models. The lexical decision and reading task response times are tested with gamma regression with inverse link. Both age of acquisition models are linear regression models, and the memory recognition accuracy results are tested with a beta regression model. These models are presented in Table 4.

For both the auditory lexical decision and reading aloud tasks, high surprisal words were associated with decreased accuracy and slower response times. In the auditory lexical decision task, the models revealed that high surprisal significantly decreased accuracy ($\beta = -2.833$, $p < .005$) and increased response times ($\beta = -3.056$, $p < .005$), while iconicity was associated with improved accuracy ($\beta = 3.036$, $p < .005$) and faster response times ($\beta = 2.152$, $p < .05$). In the read-aloud task, similar patterns were observed: high surprisal was associated with decreased accuracy ($\beta = -6.438$, $p < .001$) and lengthened response times ($\beta = 9.515$, $p < .001$), while increased iconicity improved accuracy ($\beta = 16.687$, $p < .001$) and shortened response times ($\beta = 11.895$, $p < .001$).

In the age of acquisition experiments, the Kuperman [71] model showed a strong positive association between surprisal and age of acquisition ($\beta = 8.793$, $p < .001$) and a significant negative correlation for iconicity ($\beta = -20.256$, $p < .001$). The

**Table 3. Results of the multiple linear regression model examining the influence of various linguistic factors on average surprisal. PoS refers to parts of speech. Asterisks denote statistical significance (\* p<0.05, \*\* p<0.01, \*\*\* p<0.001).**

| Model | Value |
|---|---|
| Intercept | 33.456\*\*\* |
| F statistic | 66 |
| Degrees Freedom | 15:13288 |
| Adjusted R² | 0.068 |
| length | -0.085 |
| iconicity | 9.600\*\*\* |
| length:iconicity | -6.567\*\*\* |
| n_morph | 9.527\*\*\* |
| PoS_Adverb | -3.625\*\*\* |
| PoS_Article | -3.943\*\*\* |
| PoS_Conjunction | -1.679 |
| PoS_Determiner | -3.035\*\* |
| PoS_Interjection | 5.136\*\*\* |
| PoS_Name | -0.042 |
| PoS_Noun | 5.818\*\*\* |
| PoS_Number | -2.858\*\* |
| PoS_Preposiiton | -3.675\*\*\* |
| PoS_Pronoun | -5.542\*\*\* |
| PoS_Verb | -3.203\*\* |

Glasgow Norms [72] model confirmed these trends, with surprisal positively ($\beta=6.624$, $p<.001$) and iconicity negatively correlated with age of acquisition ($\beta=-6.892$, $p<.001$). For the memory recognition experiment, both iconicity and surprisal were positively associated with improved memory accuracy, with significant effects for surprisal ($\beta=8.21$, $p<.001$) and iconicity ($\beta=3.291$, $p<.001$), confirming the enhanced memory performance for these word types.

## 4 Discussion

The present study provides insights into the relationship between surprisal, iconicity, cognitive processing, and lexeme evolution. It reveals that longer words tend to convey less information per phoneme, yet the impact of length on surprisal decreases in iconic words which are processed more effortlessly. High-information words lead to slower responses and more errors but are more likely to be remembered, while iconic words are processed faster with fewer errors, despite containing more information. These findings highlight the adaptive nature of language in response to cognitive demands, with implications for both processing and evolution. Firstly, it is evident that the length of words (i.e., number of phonemes in a word) significantly influences the distribution of information within a word such that longer words generally convey less information per phoneme than shorter words. However, interestingly, the influence of length on surprisal is almost entirely accounted for by the interaction effect between length and iconicity. Indeed, the study reveals that iconic words—words characterized by their inherent resemblance to their meaning—exhibit a dampened effect of length on information expression, which speaks to the known processing advantage of iconic words. Furthermore, bigram position in words significantly modulates information distribution, whereby the influence of the variable of length (i.e., number of phonemes in a word) on surprisal is particularly evident at word boundaries and is much weaker—or perhaps even non-existent—in the middle of longer words. Regarding psycholinguistic tasks, highly surprising words are associated with decreased accuracy and prolonged response times, while iconic words demonstrate the opposite pattern, underscoring the streamlined

**Table 4. Multiple linear regression model results constructed to test how various variables influence average surprisal. Asterisks denote statistical significance (* $p < 0.05$, ** $p < 0.01$, *** $p < 0.001$). RT refers to the response times of participants in the lexical decision and reading aloud tasks which are tested with inverse link so negative numbers are associated with increased response time due to the directionality of coefficient effects.**

| Test | Lexical Decision | | Reading Task | | Age of Acquisition | | Memory |
|---|---|---|---|---|---|---|---|
| Variable | Accuracy | RT | Accuracy | RT | [71] | [72] | Accuracy |
| Intercept | 9.673*** | 83.05 *** | 21.908 *** | 169.104 *** | 44.76*** | 20.84*** | -1.699. |
| DF | 15:10324 | 15:10324 | 15:13093 | 15:13093 | 15:12499 | 14:4086 | 11:4561 |
| **Average Surprisal** | **-2.833 \*\*** | **-3.056 \*\*** | **-6.438 \*\*\*** | **-9.515 \*\*\*** | **8.79\*\*\*** | **6.62\*\*\*** | **8.21 \*\*\*** |
| **Iconicity** | **3.036 \*\*** | **2.152 \*** | **16.687 \*\*\*** | **11.895 \*\*\*** | **-20.26\*\*\*** | **-6.89\*\*\*** | **3.291 \*\*\*** |
| Length | 6.501 *** | -13.21 *** | -13.348 *** | -57.422 *** | 33.40*** | 21.38*** | -2.468 *** |
| Morphemes | 2.142 * | -2.082 * | 5.625 *** | 4.733 *** | 1.42 | 0.17 | -3.883 *** |
| PoS_Adverb | 1.364 | 0.628 | 3.069 ** | 5.787 *** | -11.47*** | -3.60*** | -1.141 *** |
| PoS_Article | -3.059 ** | -1.374 | 0.067 | 0.036 | -2.57* | -1.54 | |
| PoS_Conjunction | -0.954 | -1.025 | 0.256 | 1.203 | -5.09*** | | |
| PoS_Determiner | -0.093 | -0.396 | 1.719 | 1.821 | -5.95*** | -2.35* | |
| PoS_Interjection | -0.198 | -2.86 ** | -1.28 | 0.025 | -3.81*** | -2.54* | 0.235 |
| PoS_Name | 0.78 | -2.506 * | 0.891 | 0.449 | -0.80 | -1.01 | 2.375 |
| PoS_Noun | 2.747 ** | -0.72 | 5.012 *** | 3.351 *** | -8.40*** | -2.24* | 5.424 *** |
| PoS_Number | 0.489 | -0.822 | 1.868 | 1.682 | -8.48*** | -2.44* | 0.443 |
| PoS_Preposiiton | -1.259 | -0.755 | 1.764 | 1.676 | -6.53*** | -2.71** | -0.98 |
| PoS_Pronoun | 1.22 | 0.037 | 2.325 * | 4.418 *** | -10.06*** | -1.97* | |
| PoS_Verb | 2.172 * | -2.974 ** | 4.581 *** | 0.242 | 1.16 | 0.37 | -10.23 |

processing and reduced cognitive load associated with iconicity. Increased surprisal and iconicity both correlate with improved recognition memory, although we consider the possibility that the influence of iconicity on memory recognition (see [10]) might be entirely explained by phonological markedness and suggest models that extend beyond bigram surprisal to test this. Finally, the study illuminates an interesting Age-of-Acquisition patterns, revealing a positive correlation between informativity and Age-of-Acquisition, and a negative correlation between iconicity and Age-of-Acquisition, suggesting differential cognitive processing and developmental trajectories for lexemes based on their linguistic properties.

In this study, we explore the concept of surprisal within the framework of information theory, focusing on how surprising phoneme sequences impact cognitive processing and lexeme evolution. Surprisal, as we define it, is intricately linked to the frequency of phoneme occurrences, where less frequent (and thus more surprising) sequences are considered phonotactically unusual or unentrenched. Building on existing research [e.g., 74–77], our study first quantifies phonotactic unusualness and then it discusses how phonological patterns and cognitive processing intersect. For example, research on the role of phonotactic variability and its impact on phonological patterns provides a crucial backdrop for understanding how surprisal might function within broader phonetic and phonological systems [78]. Similarly, contributions to the theory of probability- or exemplar-based phonology offer a framework for considering how individual variations in phoneme sequences can accumulate and affect language evolution over time [77,79]. By adopting an information-theoretic perspective, we have been able to quantify phonemic unexpectedness and assess its influence on cognitive processing.

Our findings align with both the *Lossy Context Surprisal Model* [63] and *Cognitive Load Theory* [11], offering a nuanced perspective on how phonotactic surprisal interacts with processing effort and memory. The Lossy Context Surprisal Model posits that words with higher surprisal require greater cognitive effort because they are more difficult to integrate into lossy, compressed memory representations of linguistic context. This aligns with our findings that words composed of phonotactically unentrenched (i.e., high-surprisal) sequences are processed more slowly and with lower accuracy in

lexical decision and reading tasks. However, these same words show an advantage in memory retention, consistent with the model's prediction that greater processing effort can lead to deeper encoding. Cognitive Load Theory, which describes how working memory is taxed by novel or unexpected stimuli, further supports this interpretation. High-surprisal phoneme sequences impose greater processing demands, resulting in slower reaction times and later acquisition, yet they also enhance memorability, suggesting a trade-off between processing efficiency and long-term retention. This trade-off mirrors similar findings in sentence-level processing, where unexpected words require greater attentional resources but are remembered more effectively. Thus, our results suggest that phonotactic surprisal operates within a general cognitive framework that balances processing economy with memory benefits, reinforcing the interplay between predictability, cognitive effort, and linguistic evolution.

The relationship between word length, iconicity, and surprisal provides compelling insights into the adaptive nature of lexeme evolution in response to the cognitive demands of speakers. The observed inverse relationship between word length and informativity underscores a fundamental principle of linguistic efficiency: longer words tend to convey less information per phoneme. Combined with our findings that surprisal increases cognitive load, this phenomenon suggests a cognitive economy principle at play, wherein languages evolve to balance communicative effectiveness with cognitive processing demands. Importantly, however, iconic words behave very differently because they carry more information than arbitrary words yet are processed with greater accuracy and speed. This phenomenon suggests a unique cognitive advantage conferred by iconicity, so that the intuitive connection between form and meaning expedites processing, at least partially counteracting the anticipated rise in cognitive load linked to longer or more surprising words. Consequently, the incorporation of iconic words into language not only enhances communicative efficiency and optimizes cognitive processing but also plays a pivotal role in shaping lexeme evolution, as the preference for iconicity reflects a tendency to streamline communication and adapt to the cognitive needs of speakers. In addition to the cognitive outcomes of this study, our findings on surprisal in iconic words indeed lay important groundwork for future research into the communicative utility of iconicity. By revealing how patterns of surprisal correlate with the processing of iconic forms, we open new avenues for exploring how iconicity enhances communicative effectiveness. This understanding can pave the way for investigating how iconicity functions across different linguistic contexts and its implications for language development and evolution. As we continue to examine these relationships, we can better appreciate the multifaceted roles that iconic words play in facilitating not only comprehension but also expressive communication.

Cognitive load [11] is also directly related to predictability, and this relationship can be understood within the broader framework of a cognitive economy, which suggests that language is structured to minimize cognitive effort. The theory of cognitive economy posits that human cognition is optimized to balance the processing demands of linguistic input with the need for efficient communication. As the brain constantly generates predictions about upcoming linguistic input based on prior context, words with lower predictability or higher surprisal require more cognitive resources to integrate into ongoing discourse. This increased processing effort can result in longer reading times, slower response times, and lower accuracy in tasks such as visual or auditory lexical decision tasks, as more cognitive resources are needed to process unexpected linguistic events. However, the cognitive economy principle also suggests that this additional cognitive effort may not always be detrimental. For instance, while less predictable or high surprisal words demand more processing resources, they may also benefit from deeper encoding and enhanced memory recognition. The *lossy-context surprisal model* [63] supports this idea, positing that the difficulty in processing high-surprisal words can lead to stronger memory retention due to the extra cognitive effort required. This aligns with the cognitive economy framework, as the additional investment in processing unpredictable words results in long-term benefits, such as improved memorability. The findings of the present study support this, revealing a positive correlation between memory recognition and surprisal, underscoring how cognitive resources are not just taxed but strategically allocated to maximize linguistic efficiency and retention.

The findings of our study highlight the significant influence of both surprisal and iconicity on the age at which words are learned. With the age of acquisition scores functioning as an indication of when words are learned, our results show that

words with higher surprisal tend to be acquired later, while more iconic words are learned earlier. While the relationship between iconicity and age of acquisition has been reasonably well explored (see discussion [50]), less is known about the relationship between surprisal and age of acquisition. While cognitive load is likely an important factor, another possible explanation for the delayed acquisition of highly informative words is their composition of articulatorily challenging sound sequences. For instance, consider the consonant cluster /pg/ in *upgrade* which involves a rapid transition from a voiceless plosive at the lips (for the /p/ sound) into a voiced plosive at the velum (for the /g/ sound). This articulatory complexity is likely why /pg/ is a relatively uncommon sequence in English, thereby contributing to its high Surprisal (Surprisal = 13.05), which far exceeds the average for bigrams in American English ($M = 4.16$, $SD = 1.94$). Hence, the relationship observed between surprisal, and age of acquisition may be attributed not only to cognitive development but also motor development.

The /pg/ combination in *upgrade* may also demonstrates why increased morphological complexity is associated with increased surprisal. A cursory inspection suggests that the bigrams that express the greatest amount of information typically consist of two consonants at a morpheme boundary that are unattested within morphemes (such as /pg/). Although further research in this area is recommended, this finding aligns with existing morpho-phonotactic research that shows that morphological complexity is associated with rare or unattested sequences of speech sounds that may be used to signal morpheme boundaries [e.g., 80,81].

The exploration of how bigram position within a word impacts surprisal shows that the effect of word length on predictability is most prominent at both the beginnings and ends of words. As words get longer, both the initial and final segments become more predictable, a phenomenon that suggests an adaptive strategy to reduce cognitive load. This aligns with findings [12] that early segments in less-probable words carry greater disambiguating information, helping listeners quickly narrow down lexical candidates. However, this study further shows that the predictability of word-final segments also decreases with word length, suggesting that the entire structure of the word—both at its start and end—evolves to aid efficient processing. This increasing predictability at word boundaries as word length increases may reflect an optimization strategy that allows listeners to conserve cognitive resources when processing longer, more complex words. By making both the beginning and end of longer words more predictable, the language system ensures that the listener can focus processing effort on the less predictable, information-dense middle portions of the word. The distribution of surprisal at word boundaries underscores the interplay between linguistic structure and cognitive economy, where predictability at the edges of longer words supports efficient communication and processing.

This aligns with previous research on morphonotactics [e.g., 82], which proposes that high-surprisal sequences at morpheme boundaries may serve as reliable segmentation cues in lexical processing. Rather than being a processing disadvantage, these sequences may facilitate morpheme recognition by signaling structural divisions within words. While our study does not explicitly test this hypothesis, future research could further investigate the interaction between word length, morphological complexity, and surprisal, examining whether these factors jointly influence processing efficiency and lexical representation.

We acknowledge, of course, the limitation in the analysis, as the study only focuses on English words. Whether or not these patterns are exhibited in other languages remains an open question that warrants further investigation. Future research could expand the scope of inquiry to include languages from diverse linguistic families and cultural backgrounds. Comparing the findings across languages would not only enhance our understanding of universality versus language-specific effects but also shed light on the broader principles underlying human language processing and evolution. Furthermore, average phonemic bigram surprisal—as calculated in the present study—is rather limited in that it does not consider probability beyond word boundaries. Future research might focus on more complex or complete ways of calculating surprisal.

In conclusion, this study demonstrates that—at least in the case of American English—lexemes evolve to reduce cognitive load. Longer words tend to carry less probabilistic information, particularly at word boundaries. However, when words

are processed more effortlessly, the impact of word length on surprisal decreases. High-information words lead to slower responses and more errors but are also more likely to be remembered. In contrast, iconic words are processed faster and with fewer errors, despite, stochastically, being more informative than abstract words. These findings speak to both the processing of language and lexeme evolution.

## Author contributions

**Conceptualization:** Alexander Kilpatrick, Rikke L. Bundgaard-Nielsen.

**Data curation:** Alexander Kilpatrick.

**Formal analysis:** Alexander Kilpatrick.

**Funding acquisition:** Alexander Kilpatrick.

**Investigation:** Alexander Kilpatrick.

**Methodology:** Alexander Kilpatrick.

**Project administration:** Alexander Kilpatrick.

**Resources:** Alexander Kilpatrick.

**Software:** Alexander Kilpatrick.

**Supervision:** Alexander Kilpatrick.

**Validation:** Alexander Kilpatrick.

**Visualization:** Alexander Kilpatrick.

**Writing – original draft:** Alexander Kilpatrick, Rikke L. Bundgaard-Nielsen.

**Writing – review & editing:** Alexander Kilpatrick, Rikke L. Bundgaard-Nielsen.

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
