## [Decision Letter · Decision Letter 0]

27 Jan 2025

PONE-D-24-52994Exploring the Dynamics of Shannon’s Information and Iconicity in Language Processing and Lexeme EvolutionPLOS ONE

Dear Dr. Kilpatrick,

Thank you for submitting your manuscript to PLOS ONE. After careful consideration, we feel that it has merit but does not fully meet PLOS ONE’s publication criteria as it currently stands. Therefore, we invite you to submit a revised version of the manuscript that addresses the points raised during the review process.

 Both reviewers have provided many constructive comments, ranging from general points over technical matters to small issues of presentation, so they should be helpful in improving the paper. If all comments are carefully taken into account it may be possible to avoid a second round of reviewing, although I can't promise this.

We look forward to receiving your revised manuscript.

Kind regards,

Søren Wichmann, PhD

Academic Editor

PLOS ONE

Journal Requirements:

AK -  GRANT_NUMBER: 20K13055 Japan Society for the Promotion of Science (Tokyo, JP). https://www.jsps.go.jp/english/

The funders played no role in this study. 

Reviewers' comments:

Reviewer's Responses to Questions

**Comments to the Author**

1. Is the manuscript technically sound, and do the data support the conclusions?

Reviewer #1: Yes

Reviewer #2: Yes

2. Has the statistical analysis been performed appropriately and rigorously? 

Reviewer #1: No

Reviewer #2: Yes

3. Have the authors made all data underlying the findings in their manuscript fully available?

Reviewer #1: Yes

Reviewer #2: Yes

4. Is the manuscript presented in an intelligible fashion and written in standard English?

Reviewer #1: Yes

Reviewer #2: Yes

5. Review Comments to the Author

Reviewer #1: This is an interesting study about the interaction between surprisal (information), word length, iconicity, and several measures/proxies of how words are processed. The authors combine several English data resources and run two types of linear regression models: one in which surprisal is predicted by length and iconicity, and one in which processing is predicted by surprisal and accuracy. In both cases, a set of other covariates was controlled for. According to my reading of the manuscript, the main result is a detection of a negative interaction term of length and iconicity. While both length and iconicity are positively associated with surprisal, their interaction coefficient is negative. The positive relationship between surprisal and length is dampened in the case of iconic words.

I like how the authors have motivated their analysis and model-setup and I found the literature review and contextualization to be a particularly strong part of the manuscript (but see some suggestions below). I do have some comments on the part that describes the empirical studies as well as on the discussion/conclusion section that should be addressed in my eyes. Let me go through my suggestions and questions (in no particular order).

1. I found the notation used in the formulas to be unorthodox (please, use equation numbers!). First, the variable Surprisal is defined in 117 as -logP, but the variable is not used afterwards in the subsequent equations. In line 154, they use i to refer to surprisal. In line 128, however, i seems to be a running index (as it is done conventionally). The formatting is a bit odd here, because i should be a subscript everywhere in the formula (entropy is not defined so that the product within the brackets is multiplied by i). This needs to be cleaned up. For instance, one could use S_i as surprisal in line 117 and then sum over all S_i in line 154.

2. It is not clear to me how the authors arrived at what they refer to as “Zipf transformation” in line 139. Maybe I missed something but Brysbaert and New (2009) investigate transformations of the type log(freq + 1). Where does “number of word types” in the denominator come from and why do we add +3? Zipf’s frequency law essentially asserts that the relationship between frequency and rank follows a power law with a coefficient of around -1, or equivalently, that log(freq) and log(rank) are linearly related with a slope of -1. Where does rank surface in 139 and why is lexicon size used to normalize in addition to corpus size? This is not clear to me.

3. Sometimes, I found that terminological choices were not optimal. The authors often refer to data resources as “corpora” (e.g. line 365) although they actually mean data bases consisting of structured (psycholinguistic/linguistic) data. In the present context, I would also use “diphones/biphones” rather than “bigrams” because the latter could be misinterpreted as pairs of wordform tokens. In the caption of Figures 1 & 2, the authors mention “plots for the distribution of words” without specifying what exactly is shown. In line 440, it is said that AoA is used as “proxy for the cognitive development”, but this is not what AoA norms measure (evidently, AoA norms are properties of lexemes, and lexemes do not have a “cognitive development”).

4. In general, I think that section numbers would help a lot. The hierarchy of sections was not always clear. For example “Part One: The Master Dataset” is on the same level as “Part One: Predictions”, although the dataset described in the former dataset is relevant for part two as well. I strongly recommend structuring the manuscript in such a way, that the authors discuss all data used in the whole study in one section, then methodological setup (with subsections for both “Parts”) in another section (including predictions; or talk about the predictions already in the introduction), a section for the results, and finally the discussion.

5. I have some questions and comments about the statistical modeling procedure. First, it is not clear to me whether the beta coefficients represent normalized effect sizes. On the one hand, I assume that this is the case because the authors comment on their size but on the other hand, I do not see a z-transformation in the code. Would it maybe make sense to have such a normalization in the case of the linear models (Table 2) to be able to compare effect sizes? Also, what does the distribution of entropy look like in the dataset?

Second, the authors show that R-squared for the model with Count as dependent variable is particularly small and argue that this demonstrates that the other measures are superior with respect to their explanatory power. I’d like to object and rather say that using a linear Gaussian model together with an independent variable as skewed as Count (Figure 1) will necessarily entail a small R-squared value. But then it is not that the variable is less valuable but rather opting for a linear model in such a case is simply a bad methodological choice. In other words: just use the logarithm to de-skew the variable (or go for a generalized linear model with log transformation).

Third, I object against using linear (Gaussian) models in Part 2 (i.e., Table 3). The outcome variables used here are connected to diverse distributional families: accuracy is bound to the unit interval and has a ceiling effect (as pointed out by the authors), RT is restricted to positive values but is typically skewed to the right, and AoA is restricted to positive values but typically normally distributed. For none of these variables (except for AoA if one is far away from 0) the use of Gaussian linear models is recommended. Rather employ generalized linear models. For accuracy either use Quasibinomial models or beta regression, for RT use Gamma/exponential models with inverse link (take care of the signs when interpreting the coefficients!), and for AoA either use Gaussian linear models as before (or Gamma/exponential models, maybe with identity link), just to provide some ideas.

6. I have a very general comment on surprisal. First, I found it great that the authors investigate average surprisal per word. Essentially, this is a measure of how unorthodox phoneme sequences (of length two) are within a word. Surprisal is negatively associated with frequency, and frequency obviously is a measure of how common things are. Words with high surprisal hence are phonotactically strange and words with low surprisal are phonotactically familiar.

Now, what I was wondering throughout was what the manuscript would look like if we would simply replace “informative/surprising” with the terms “rare”, “(phonotactically) unfamiliar”, or “unentrenched”. I could write a paper that has an entirely different theoretical framing about the relationship between “phonotactic entrenchment, iconicity, and length” by drawing on exactly the same data and statistical analysis. So, my question is: how specific are the findings to the information theoretic approach adopted here? Or could everything be interpreted, say, in the light of entrenchment as well? I think the authors should comment on this in the discussion section (cf related work by Wedel and Pierrehumbert).

7. I found one side-result to be particularly interesting (also because it relates to what I have been working on a couple of years ago), namely the positive effect of the number of morphemes on surprisal. Morphological complexity entails phonotactically odd sequences. Interestingly, research on morphonotactics (in particular by Nikolaus Ritt, WU Dressler, and Kasia Dziubalska-Kolaczyk) even suggests that such odd sequences are preserved because they help in the decomposition of words into morphemes, and their oddness makes them reliable boundary signals. That is, high-surprisal sequences in morphologically complex words are hypothesized to be a feature rather than a bug. It would be interesting to see if this effect remains when controlling for word length (in an interaction term), but maybe not as part of this analysis. The authors could comment on the implications of their findings for research on (mor)phonotactics a little more in the discussion to provide some more contextualization.

8. Linked to the topic of morpheme boundaries, I wonder why the authors have excluded diphones that span word boundaries. I guess that this was a pragmatic decision because (most of) the databases used here show information on the word level. On the other hand, their corpus data would make it possible to also investigate boundary spanning items. I do not say that the authors should change their datasets, but this limitation needs to be commended on in the conclusion section.

9. The authors highlight the evolutionary dimension of their approach at several points in the manuscript. While I do see what they are aiming at, it can always be dangerous to make evolutionary claims based on synchronic data. I think, the diachronic processes involved here should be discussed in more detail. At the very least, the authors should move the diachronic English example from the discussion section to the introduction and provide some additional examples. Then, they could argue that the current synchronic state of the linguistic system could be seen as a point where the system is attracted to (could it?), hence enabling claims about evolutionary processes. As presented now, however, a bit of more context is missing.

10. Finally, I found that the conclusion section is a bit detached from the literature. The authors take their time to discuss the results (which is good), but do not provide much contextualization. I have made some suggestions above, and I encourage the authors to add some additional links to extant research.

To summarize, I thought that this is a really interesting contribution but there are still some issues to be dealt with. Most of what I have discussed above, however, can be accounted for without a huge amount of effort of rewriting the entire manuscript.

Minor things:

250: mere > more

355: was > were

491: “effect of length … is most apparent at the beginning and ends”: first, “beginning and end”; second, one could also say that it is most apparent in the middle; actually, it is apparent from the contrast between the middle and the boundaries (beginning & end), I’d say

510: modelling > model

547: in above > above

Reviewer #2: I thought this was an outstanding paper. The study analyzes data from pre-existing datasets to investigate the relationship between different measures of information (as bigram suprisal), iconicity (form-meaning resemblance), and word length in English words, and then tests these variables as predictors of psycholinguistic experimental findings related to cognitive processing and learning. A key finding is that iconic words resist the typical relationship in which longer words are, on average, more predictable to facilitate processing – hypothetically because iconicity compensates to help processing of more surprising lengthy words. The paper puts it all together into a compelling framework of lexeme evolution.

With the caveat that I am not especially sophisticated in my knowledge of information theory and the calculation of information metrics, I found the analyses to be clearly explained, logical, and carefully done. The paper is clearly written, and the methods and results are presented clearly. In my view, the paper is nearly publishable in its current form, although I have some very minor suggestions/notes for the authors to consider:

The abstract puts emphasis on Cognitive Load Theory and the Lossy Context Surprisal Model, but then the paper does not return to these after the intro and literature review. It may be worth drawing this connection more explicitly in the discussion.

One paper that the authors might find helpful is:

Dingemanse, Mark, and Bill Thompson. 2020. ‘Playful Iconicity: Structural Markedness Underlies the Relation between Funniness and Iconicity’. Language and Cognition 12 (1): 203–24. https://doi.org/10.1017/langcog.2019.49.

This paper includes an analysis of iconicity in relation to structural markedness based on measures of log letter frequency, but also looked at measures of phonological density, biphone probability, and triphone probability. Also the already cited Winter et al 2024 (BRM), which reports the iconicity ratings used in the current study, also report the correlation with log letter frequency.

Finally, a more tentative thought … in interpreting and discussing the results, the authors may consider putting more emphasis on iconicity as a semiotic mode of depiction and why iconic words might defy predictable phoneme sequences for the purpose of more accurately depicting meaning. This point seems a little lost, to my ear, in the paper’s emphasis on iconicity as easing processing. In this regard, the paper might need to more carefully tease apart causation and correlation in their model of lexeme evolution.

Minor line-by-line comments:

Ln 36. “Iconic words like buzz are advantageous in cognitive processing, as their form-meaning congruence enhances recognition and recall [6,7,8].” Are there references that more directly support this point?

Ln 167. “Some have suggested that probabilistic reduction reallocates attentional or cognitive resources, leading to a trade-off with complexity in acoustic signal [31], the present study suggests a perceptual corollary to this behaviour.”

This sentence seems disjointed.

Ln 211 e.g., WHAAT? [wæːt])”. Is this meant to be the question word “what?” I don’t recognize this as an iconic English word.

Figures 1 and 2. I think it would be helpful to include labels for axes and units, and perhaps a little more guidance in the figure caption.

Ln 473. “we three simple” – missing verb

Ln 478 .“Similarly, the Word End model revealed F(1, 13302) = 596.7, β = -0.171, p < .001, R2 = 0.043.” I would explain in words, not numbers, what it “revealed”

Ln 677. “Our results reveal a negative correlation between surprisal and age of acquisition and a positive correlation between iconicity and age of acquisition.”

I think this is backwards? My personal preference would be within the discussion to describe this in more explicit, functional terms, e.g., “more iconic words are learned earlier”

Ln 695. “highly frequent words like the would lose their acoustic”. Mark ‘the’ with quotes or italics or something. I think I saw this in other spots in the paper earlier, come to thin of it.

The authors might double-check the information of a few citations, including 19 and 61.

6. PLOS authors have the option to publish the peer review history of their article (what does this mean? ). If published, this will include your full peer review and any attached files.

**Do you want your identity to be public for this peer review?** For information about this choice, including consent withdrawal, please see our Privacy Policy .

Reviewer #1: **Yes: ** Andreas Baumann

Reviewer #2: No

---

## [Author Response · Author response to Decision Letter 1]

28 Feb 2025

Dear Editor,

We sincerely appreciate the opportunity to revise our manuscript, and we are grateful to the reviewers for their insightful comments and suggestions. Their feedback has significantly improved the clarity, structure, and methodological rigor of our study.

In response to the reviewers’ concerns, we have carefully addressed each point. Among the key changes, we have:

• Modified the statistical analyses in accordance with Reviewer 1.

• Included additional paragraphs of our theoretical frameworks in the discussion section.

• Moved the discussion of lexeme evolution to the literature review.

We believe these revisions have strengthened the manuscript, and we appreciate the reviewers’ constructive feedback, which has led to meaningful improvements. Please find our detailed response attached. All modification both in this document and the main manuscript, are highlighted except where it involves simple deletion.

Thank you for your time and consideration. We look forward to your feedback.

Sincerely,

Alexander Kilpatrick

Reviewer #1:

I found the notation used in the formulas to be unorthodox (please, use equation numbers!). First, the variable Surprisal is defined in 117 as -logP, but the variable is not used afterwards in the subsequent equations. In line 154, they use i to refer to surprisal. In line 128, however, i seems to be a running index (as it is done conventionally). The formatting is a bit odd here, because i should be a subscript everywhere in the formula (entropy is not defined so that the product within the brackets is multiplied by i). This needs to be cleaned up. For instance, one could use S_i as surprisal in line 117 and then sum over all S_i in line 154.

Thank you for your thoughtful feedback. I have addressed this issue by reiterating the surprisal formula in the definition of average surprisal to maintain clarity and consistency.

2. It is not clear to me how the authors arrived at what they refer to as “Zipf transformation” in line 139. Maybe I missed something but Brysbaert and New (2009) investigate transformations of the type log(freq + 1). Where does “number of word types” in the denominator come from and why do we add +3? Zipf’s frequency law essentially asserts that the relationship between frequency and rank follows a power law with a coefficient of around -1, or equivalently, that log(freq) and log(rank) are linearly related with a slope of -1. Where does rank surface in 139 and why is lexicon size used to normalize in addition to corpus size? This is not clear to me.

This is no longer applicable as we no longer examine Surprisal calculated on alternate frequency metrics.

Sometimes, I found that terminological choices were not optimal. The authors often refer to data resources as “corpora” (e.g. line 365) although they actually mean data bases consisting of structured (psycholinguistic/linguistic) data.

Instances of “Corpora” have been changed to “datasets” where applicable.

In the present context, I would also use “diphones/biphones” rather than “bigrams” because the latter could be misinterpreted as pairs of wordform tokens.

We appreciate the reviewer’s suggestion. However, we have chosen to retain the term "bigram" as it originates from information theory and is widely used in computational linguistics to refer to sequential units at various levels, including phonemes. We define "phonemic bigrams" explicitly in the first paragraph of the introduction to clarify that we are referring to pairs of phonemes rather than wordform tokens. Given this definition, we believe there is no risk of misinterpretation, and the term remains the most appropriate for our study’s methodological framework.

In the caption of Figures 1 & 2, the authors mention “plots for the distribution of words” without specifying what exactly is shown.

These plots are no longer featured in the manuscript.

In line 440, it is said that AoA is used as “proxy for the cognitive development”, but this is not what AoA norms measure (evidently, AoA norms are properties of lexemes, and lexemes do not have a “cognitive development”).

We appreciate the reviewer’s clarification. To address this concern, we have revised the text to remove any implication that Age of Acquisition (AoA) norms directly measure cognitive development. Instead, we now describe AoA as reflecting the relative difficulty of words in early language learning, which aligns more precisely with what these norms capture. This revision ensures that our interpretation remains consistent with the established understanding of AoA as a property of lexemes rather than a measure of individual cognitive development.

In general, I think that section numbers would help a lot. The hierarchy of sections was not always clear. For example “Part One: The Master Dataset” is on the same level as “Part One: Predictions”, although the dataset described in the former dataset is relevant for part two as well. I strongly recommend structuring the manuscript in such a way, that the authors discuss all data used in the whole study in one section, then methodological setup (with subsections for both “Parts”) in another section (including predictions; or talk about the predictions already in the introduction), a section for the results, and finally the discussion.

We appreciate the reviewer’s suggestion regarding the manuscript’s structure. In response, we have added section numbers to clarify the hierarchy and improve readability. Additionally, we have restructured the manuscript to ensure that all data is discussed in a dedicated section before outlining the methodological setup, predictions, and results. We believe these changes enhance the overall clarity and coherence of the study. Thank you for this valuable recommendation.

I have some questions and comments about the statistical modeling procedure. First, it is not clear to me whether the beta coefficients represent normalized effect sizes. On the one hand, I assume that this is the case because the authors comment on their size but on the other hand, I do not see a z-transformation in the code. Would it maybe make sense to have such a normalization in the case of the linear models (Table 2) to be able to compare effect sizes? Also, what does the distribution of entropy look like in the dataset?

We appreciate the reviewer’s detailed comments regarding our statistical modelling choices. In response to this feedback, we have removed the analysis involving the different methods for calculating surprisal from the manuscript. Upon further consideration, we found that this analysis did not meaningfully contribute to the central findings of the study and introduced unnecessary complexity. Removing it has streamlined the manuscript and improved clarity.

Additionally, we acknowledge the reviewer’s point regarding the potential benefits of normalizing predictors for comparability. However, since this specific analysis has now been omitted, the concern regarding R-squared values and transformation of skewed variables is no longer relevant to the revised manuscript.

We sincerely appreciate the reviewer’s insights, which have helped us refine our focus and improve the overall coherence of our study.

Second, the authors show that R-squared for the model with Count as dependent variable is particularly small and argue that this demonstrates that the other measures are superior with respect to their explanatory power. I’d like to object and rather say that using a linear Gaussian model together with an independent variable as skewed as Count (Figure 1) will necessarily entail a small R-squared value. But then it is not that the variable is less valuable but rather opting for a linear model in such a case is simply a bad methodological choice. In other words: just use the logarithm to de-skew the variable (or go for a generalized linear model with log transformation).

As above, this has been removed from the manuscript.

Third, I object against using linear (Gaussian) models in Part 2 (i.e., Table 3). The outcome variables used here are connected to diverse distributional families: accuracy is bound to the unit interval and has a ceiling effect (as pointed out by the authors), RT is restricted to positive values but is typically skewed to the right, and AoA is restricted to positive values but typically normally distributed. For none of these variables (except for AoA if one is far away from 0) the use of Gaussian linear models is recommended. Rather employ generalized linear models. For accuracy either use Quasibinomial models or beta regression, for RT use Gamma/exponential models with inverse link (take care of the signs when interpreting the coefficients!), and for AoA either use Gaussian linear models as before (or Gamma/exponential models, maybe with identity link), just to provide some ideas.

We now use quasibinomial regression for accuracy (to handle the ceiling effect). We apply Gamma regression with inverse link for reaction times (RT) to account for skewness and the non-negative nature of RT data. We use Beta regression for memory accuracy, as it is a proportion variable. For Age of Acquisition (AoA), we check its distribution and apply either a Gaussian linear model or a Gamma regression, depending on normality. These changes ensure that our models align better with the data distributions, improving the robustness of our findings. We thank the reviewer for these valuable suggestions, which have helped refine our analysis. The code for the new statistical tests is listed below for your convenience.

MALD_ACC_Model <- glm(LexicalD_ACC_A_MALD ~ informativity + iconicity + length + n_morph + POS_BRYS, data = MALD_Data, family = quasibinomial(link = "logit"))

MALD_RT_Model <- glm(LexicalD_RT_A_MALD ~ informativity + iconicity + length + n_morph + POS_BRYS, data = MALD_Data, family = Gamma(link = "inverse"))

ELP_ACC_Model <- glm(Naming_ACC_ELP ~ informativity + iconicity + length + n_morph + POS_BRYS, data = ELP_Data, family = quasibinomial(link = "logit"))

ELP_RT_Model <- glm(Naming_RT_ELP ~ informativity + iconicity + length + n_morph + POS_BRYS, data = ELP_Data, family = Gamma(link = "inverse"))

REC_ACC_Model <- betareg(Recog_Memory ~ informativity + iconicity + length + n_morph + POS_BRYS, data = RecMem_Data)

6. I have a very general comment on surprisal. First, I found it great that the authors investigate average surprisal per word. Essentially, this is a measure of how unorthodox phoneme sequences (of length two) are within a word. Surprisal is negatively associated with frequency, and frequency obviously is a measure of how common things are. Words with high surprisal hence are phonotactically strange and words with low surprisal are phonotactically familiar.

Now, what I was wondering throughout was what the manuscript would look like if we would simply replace “informative/surprising” with the terms “rare”, “(phonotactically) unfamiliar”, or “unentrenched”. I could write a paper that has an entirely different theoretical framing about the relationship between “phonotactic entrenchment, iconicity, and length” by drawing on exactly the same data and statistical analysis. So, my question is: how specific are the findings to the information theoretic approach adopted here? Or could everything be interpreted, say, in the light of entrenchment as well? I think the authors should comment on this in the discussion section (cf related work by Wedel and Pierrehumbert).

We appreciate the reviewer’s insightful comment regarding the relationship between surprisal, phonotactic entrenchment, and frequency effects. As suggested, we have expanded the discussion to address the extent to which our findings are specific to the information-theoretic framework versus how they might also be interpreted within an entrenchment-based approach. We agree that surprisal, as we have defined it, is closely related to phonotactic familiarity, and it is possible to frame our results in terms of phonotactic entrenchment.

Indeed, prior work (e.g., Wedel, 2012; Pierrehumbert, 2001) has demonstrated that phonotactic probabilities shape lexical representation and influence how sequences become more entrenched over time through exposure. Our information-theoretic approach aligns with this perspective but provides a quantitative measure that explicitly captures cognitive processing costs associated with unexpected phoneme sequences. While entrenchment emphasizes how phonotactic familiarity emerges through repeated exposure, information theory highlights how unpredictability modulates processing difficulty and memory retention. We have now incorporated this discussion into the manuscript to clarify the broader theoretical implications of our findings. Thank you for this valuable suggestion, which has helped us refine the theoretical positioning of our work.

I found one side-result to be particularly interesting (also because it relates to what I have been working on a couple of years ago), namely the positive effect of the number of morphemes on surprisal. Morphological complexity entails phonotactically odd sequences. Interestingly, research on morphonotactics (in particular by Nikolaus Ritt, WU Dressler, and Kasia Dziubalska-Kolaczyk) even suggests that such odd sequences are preserved because they help in the decomposition of words into morphemes, and their oddness makes them reliable boundary signals. That is, high-surprisal sequences in morphologically complex words are hypothesized to be a feature rather than a bug. It would be interesting to see if this effect remains when controlling for word length (in an interaction term), but maybe not as part of this analysis. The authors could comment on the implications of their findings for research on (mor)phonotactics a little more in the discussion to provide some more contextualization.

Thank you for your insightful comment. We have expanded the discussion to address the relationship between morphological complexity, phonotactically odd sequences, and surprisal, as you suggested. We appreciate your suggestion and have added this context to better situate our findings within morphonotactic research.

Linked to the topic of morpheme boundaries, I wonder why the authors have excluded diphones that span word boundaries. I guess that this was a pragmatic decision because (most of) the databases used here show information on the word level. On the other hand, their corpus data would make it possible to also investigate boundary spanning items. I do not say that the authors should change their datasets, but this limitation needs to be commended on in the conclusion section.

As the reviewer correctly noted, the decision to focus on word-level bigrams was primarily pragmatic, given the structure of the datasets we used. While the corpus data could indeed support an investigation into boundary-spanning items, we felt that such an analysis would require a different approach to account for additional complexities beyond the word boundary. We have acknowledged this limitation in the conclusion and highlighted the potential for future research to explore more complex ways of calculating surprisal, including considering diphones across word boundaries.

The authors highlight the evolutionary dimension of their approach at several points in the manuscript. While I do see what they are aiming at, it can always be dangerous to make evolutionary claims based on synchronic data. I think, the diachronic processes involved here should be discussed in more detail. At the very least, the authors should move the diachronic English example from the discussion section to the introduction and provide some additional examples. Then, they could argue that the current synchronic state of the linguistic system could be seen as a point where the system is attracted to (could it?), hence enabling claims about evolutionary processes. As presented now, however, a bit of more context is missing.

Thank you for your valuable feedback. We agree that evolutionary claims based on synchronic data should be handled carefully, and we appreciate your suggestion to provide more context on the diachronic processes involved. In response, we have moved the diachronic English example from the discussion to the introduction and added additional examples to clarify our p

---

## [Editor Report · Decision Letter 1]

4 Mar 2025

Exploring the Dynamics of Shannon’s Information and Iconicity in Language Processing and Lexeme Evolution

PONE-D-24-52994R1

Dear Dr. Kilpatrick,

We’re pleased to inform you that your manuscript has been judged scientifically suitable for publication and will be formally accepted for publication once it meets all outstanding technical requirements.

Kind regards,

Søren Wichmann, PhD

Academic Editor

PLOS ONE
---

## [Editor Report · Acceptance letter]

PONE-D-24-52994R1

PLOS ONE

Dear Dr. Kilpatrick,

I'm pleased to inform you that your manuscript has been deemed suitable for publication in PLOS ONE. Congratulations! Your manuscript is now being handed over to our production team.

Kind regards,

on behalf of

Dr. Søren Wichmann

Academic Editor

PLOS ONE